# MADNet: Maximizing Addressee Deduction Expectation for Multi-Party Conversation Generation

**Jia-Chen Gu**[1][*] **Chao-Hong Tan**[1][*] **Caiyuan Chu**[2], **Zhen-Hua Ling**[1][†],
**Chongyang Tao**[3], **Quan Liu**[2,4], **Cong Liu**[1,2]

[1]National Engineering Research Center of Speech and Language Information Processing,
University of Science and Technology of China, Hefei, China
[2]iFLYTEK Research, Hefei, China
[3]Peking University, Beijing, China
[4]State Key Laboratory of Cognitive Intelligence
{gujc,zhling}@ustc.edu.cn, chtan@mail.ustc.edu.cn,
chongyangtao@pku.edu.cn, {cychu2,quanliu,congliu2}@iflytek.com

## Abstract

Modeling multi-party conversations (MPCs) with graph neural networks has been proven effective at capturing complicated and graphical information flows. However, existing methods rely heavily on the necessary addressee labels and can only be applied to an ideal setting where each utterance must be tagged with an "@" or other equivalent addressee label. To study the scarcity of addressee labels which is a common issue in MPCs, we propose MADNet that **m**aximizes **a**ddressee **d**eduction expectation in heterogeneous graph neural networks for MPC generation. Given an MPC with a few addressee labels missing, existing methods fail to build a consecutively connected conversation graph, but only a few separate conversation fragments instead. To ensure message passing between these conversation fragments, four additional types of latent edges are designed to complete a fully-connected graph. Besides, to optimize the edge-type-dependent message passing for those utterances without addressee labels, an Expectation-Maximization-based method that iteratively generates silver addressee labels (E step), and optimizes the quality of generated responses (M step), is designed. Experimental results on two Ubuntu IRC channel benchmarks show that MADNet outperforms various baseline models on the task of MPC generation, especially under the more common and challenging setting where part of addressee labels are missing.

## 1 Introduction

The development of intelligent dialogue systems that are able to engage in conversations with humans, has been one of the longest running goals in artificial intelligence (Kepuska and Bohouta, 2018; Berdasco et al., 2019; Zhou et al.,

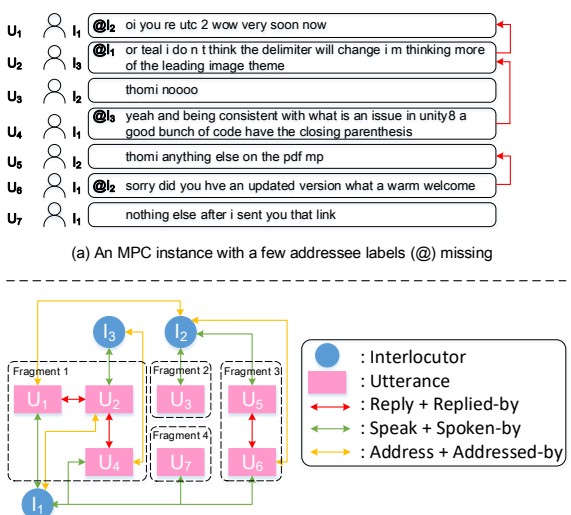

(a) An MPC instance with a few addressee labels (@) missing

(b) The graphical information flow and fragments established in HeterMPC

Figure 1: (a) A randomly sampled MPC instance from the Ubuntu IRC dataset (Ouchi and Tsuboi, 2016) where a few addressee labels, i.e., "@", are missing. (b) Illustration of the graphical information flow and conversation fragments of the instance above established in HeterMPC (Gu et al., 2022). Here, the bidirectional edges are merged for simplicity.

2020). Thanks to breakthroughs in sequence modeling (Sutskever et al., 2014; Vaswani et al., 2017) and pre-trained language models (PLMs) (Radford et al., 2019; Devlin et al., 2019; Lewis et al., 2020), researchers have proposed various effective models for conversations between two participants (Serban et al., 2016; Wen et al., 2017; Zhang et al., 2020). Recently, researchers have paid more attention to a more practical and challenging scenario involving more than two participants, which is well known as multi-party conversations (MPCs) (Ouchi and Tsuboi, 2016; Zhang et al., 2018; Le et al., 2019; Hu et al., 2019; Wang et al., 2020; Gu et al., 2021, 2022). Unlike two-party conversations, utterances in an MPC can be spoken by anyone and address anyone else in this conversation, constituting a *graphical* information flow.

---

[*]Equal contribution.
[†]Corresponding author.

Encoding MPC contexts with either homogeneous (Hu et al., 2019) or heterogeneous (Gu et al., 2022) graph neural networks (GNNs) has been proven effective at modeling graphical information flows. These methods rely heavily on the necessary addressee labels and can only be applied to an ideal setting where each utterance must be tagged with an "@" or other equivalent addressee label, to establish a consecutively connected graph. However, interlocutors in MPCs do not always strictly obey the talking rule of specifying their addressees in each utterance, as shown by a randomly sampled MPC instance in Figure 1(a). Statistics show that addressees of 55% of the utterances in the Ubuntu IRC dataset (Ouchi and Tsuboi, 2016) are not specified. In this common case, the expected conversation graph in previous work became fragmented (Hu et al., 2019; Gu et al., 2022) as shown in Figure 1(b). Therefore, nodes without direct connections cannot exchange information between each other through one-hop message passing. Despite disconnected nodes can instead be accessed indirectly via other detours through multi-hop passing, inevitable information loss and passing latency will affect generation performance significantly. But this common issue has not been studied in previous work.

In light of the above issues, we propose MADNet that **m**aximizes **a**ddressee **d**eduction expectation in heterogeneous graph neural networks to mitigate performance degradation and to enhance model robustness, for MPC generation conditioned on incomplete addressee labels. Given an MPC with a few addressee labels missing, existing methods fail to build a consecutively connected conversation graph, but only a few separate conversation fragments instead (Hu et al., 2019; Gu et al., 2022). To ensure message passing between these conversation fragments, four additional types of latent edges are designed to complete a fully-connected graph. In this way, nodes without direct connections can also directly interact with each other, and be distinguished from existing edges by parameterization. Furthermore, edge-type-dependent message passing has been verified effective at MPC modeling (Gu et al., 2022). In order to optimize edge characterization and message passing for utterances without addressee labels, a hard Expectation-Maximization-based approach (Brown et al., 1993; Shen et al., 2019; Min et al., 2019) is designed for addressee deduction. On the one hand,

the expectation steps iteratively generate silver addressee labels by considering the addressee of an utterance as a discrete latent variable. On the other hand, the maximization steps selects the addressee with the highest probability from the addressee distribution and optimize the generative dialogue model. As the number of EM iterations increases, the accuracy of the latent addressee distribution as well as the quality of generated responses can be improved simultaneously. Compared with previous methods, MADNet can be applied to more common and challenging MPC scenarios, indicating its generalization and robustness.

To measure the effectiveness of the proposed method, we evaluate the performance on two benchmarks based on Ubuntu IRC channel. One was released by Ouchi and Tsuboi (2016) where a few addressee labels were missing. The other was released by Hu et al. (2019) where addressee labels were provided for each utterance. Experimental results show that MADNet outperforms previous methods by significant margins, achieving a new state-of-the-art performance for MPC generation especially in the more common and challenging setting where a few addressee labels are missing.

In summary, our contributions in this paper are three-fold: 1) To the best of our knowledge, this paper makes the first attempt to explore the issue of missing addressee labels and to target on more common MPC scenarios. 2) A fully-connected heterogeneous graph architecture along with EM training is designed to help deduce the addressee of an utterance for improving MPC generation. 3) Experimental results show that our proposed model achieves a new state-of-the-art performance of MPC generation conditioned on incomplete addressee labels on two Ubuntu IRC benchmarks.

## 2   Related Work

**Multi-Party Conversation**   Existing methods on building dialogue systems can be generally categorized into generation-based (Serban et al., 2016; Wen et al., 2017; Young et al., 2018; Zhang et al., 2020) or retrieval-based approaches (Wu et al., 2017; Zhou et al., 2018; Tao et al., 2019; Gu et al., 2020). In this paper, we study MPC generation, where in addition to utterances, interlocutors are also important components who play the roles of speakers or addressees. Previous methods have explored retrieval-based approaches for MPCs. For example, Ouchi and Tsuboi

(2016) proposed the dynamic model which updated speaker embeddings with conversation streams. Zhang et al. (2018) proposed speaker interaction RNN which updated speaker embeddings role-sensitively. Wang et al. (2020) proposed to track the dynamic topic in a conversation. Gu et al. (2021) proposed jointly learning "who says what to whom" in a unified framework by designing self-supervised tasks during pre-training. On the other hand, Hu et al. (2019) explored generation-based approaches by proposing a graph-structured network (GSN). At its core was an utterance-level graph-structured encoder that encoded the conversation context using homogeneous GNNs. Gu et al. (2022) proposed HeterMPC to model the complicated interactions between utterances and interlocutors with a heterogeneous graph (Sun and Han, 2012), where two types of nodes and six types of edges were designed to model heterogeneity. Contemporaneous to our work, Li and Zhao (2023) focus on predicting the missing addressee labels in pre-training instead, but still only target at the ideal setting during fine-tuning where each utterance must be tagged with the addressee label.

**Expectation-Maximization**  This algorithm is used to find local maximum likelihood parameters of a statistical model in cases where the equations cannot be solved directly (Dempster et al., 1977; Dayan and Hinton, 1997; Do and Batzoglou, 2008). Rather than sampling a latent variable from its conditional distribution, a hard EM approach which takes the value with the highest posterior probability as prediction is designed (Brown et al., 1993). This approach has been proven effective at improving the performance of various NLP tasks such as dependency parsing (Spitkovsky et al., 2010), machine translation (Shen et al., 2019), question answering (Min et al., 2019) and diverse dialogue generation (Wen et al., 2023). In this paper, we study whether considering addressees as a latent variable and deducing with hard EM is useful for modeling conversation structures and improving MPC generation performance.

Compared with GSN (Hu et al., 2019) and HeterMPC (Gu et al., 2022) that are the most relevant to this work, a main difference should be highlighted. These methods target only on an ideal setting where addressee labels of all utterances are necessary, while the proposed method is suitable for more common conversation sessions where a few addressee labels are missing. To the best of our knowledge, this paper makes the first attempt to extend to more common MPC scenarios and to explore the issue of missing addressee labels by maximizing addressee deduction expectation for MPC generation.

## 3 Preliminaries

**Problem Formulation**  The task of response generation in MPCs is to generate an appropriate response $\bar{r}$ given the conversation history, the speaker of a response, and which utterance the response is going to reply to. This can be formulated as:

$$
\begin{aligned}
\bar{r} &= \underset{r}{\arg\max} \, logP(r|\mathbb{G}, c; \theta) \\
&= \underset{r}{\arg\max} \sum_{t=1}^{|r|} logP(r_t|\mathbb{G}, c, r_{<t}; \theta).
\end{aligned}
\tag{1}
$$

Here, $\mathbb{G}$ is a heterogeneous graph, $c$ is the context of dialogue history, $\theta$ is the model parameters. The speaker and addressee of the response are known and its contents are masked. The response tokens are generated in an auto-regressive way. $r_t$ and $r_{<t}$ stand for the $t$-th token and the first $(t-1)$ tokens of response $r$ respectively. $|r|$ is the length of $r$.

Next, we briefly present the key process of the HeterMPC baseline (Gu et al., 2022) to avoid lengthy method descriptions, which shares the GNN backbone with our method. Readers can also refer to Gu et al. (2022) for more details.

**Graph Construction**  Given an MPC instance composed of $M$ utterances and $I$ interlocutors, a heterogeneous graph $\mathbb{G}(\mathbb{V}, \mathbb{E})$ is constructed. Specifically, $\mathbb{V}$ is a set of $M + I$ nodes. Each node denotes either an utterance or an interlocutor. $\mathbb{E} = \{e_{p,q}\}_{p,q=1}^{M+I}$ is a set of directed edges, where each edge $e_{p,q}$ describes the connection from node $p$ to node $q$. Six types of meta relations in HeterMPC (Gu et al., 2022) and four additional types of latent edges proposed in this paper are introduced to describe directed edges between nodes, which will be elaborated in Section 4.1. It is notable that $e_{p,q}$ is set to *NULL* if there is no connection between two nodes, so that interactions between them can only be conducted indirectly via detours through multi-hop passing.

**Node Initialization**  Each node is represented as a vector, and two strategies are designed to initialize the node representations for utterances and interlocutors respectively. Utterances are first encoded individually by stacked Transformer

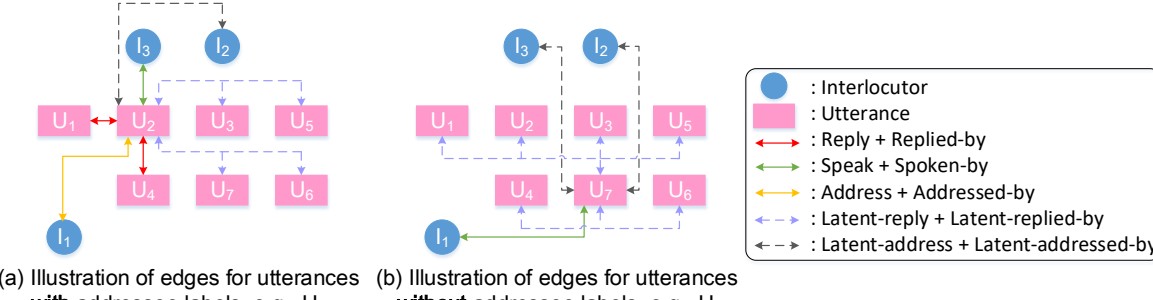

Figure 2: Illustration of edges for utterances (a) **with** and (b) **without** addressee labels respectively in a fully-connected MPC graph for the instance in Figure 1.

layers that can be initialized by PLMs, e.g. encoder of BART (Lewis et al., 2020), to derive the contextualized and utterance-level representations. On the other hand, interlocutors in a conversation are indexed according to their speaking order and the embedding vector for each interlocutor is derived by looking up an order-based interlocutor embedding table (Gu et al., 2020) that is updated during learning.

**Node Updating**   The initialized node representations are updated by feeding them into the built graph for absorbing context information. Heterogeneous attention weights between connected nodes and message passing over the graph are calculated in a node-edge-type-dependent manner. Parameters are introduced to maximize feature distribution differences for modeling heterogeneity (Schlichtkrull et al., 2018; Zhang et al., 2019; Hu et al., 2020). After collecting the information from all source nodes to a target node, a node-type-dependent feed-forward network followed by a residual connection (He et al., 2016) is employed for aggregation.   To let each utterance token also have access to graph nodes, an additional Transformer layer is placed for utterance nodes specifically.   After completing an iteration, the outputs of utterance and interlocutor nodes are employed as the inputs of the next iteration.

**Decoder**   It follows the standard implementation of Transformer decoder (Vaswani et al., 2017) for generation, and can be initialized by PLMs, e.g. decoder of BART (Lewis et al., 2020). In each decoder layer, a masked self-attention operation is performed where each token cannot attend to future tokens to avoid information leakage. Then, a cross-attention operation over the node representations output by the graph encoder is performed to incorporate graph information for decoding.

## 4   Methodology

In this section, we first describe how to construct a fully-connected conversation graph to ensure message passing between conversation fragments. Then, the expectation and maximization steps for addressee deduction are defined. Finally, an addressee initialization method is designed for better convergence of the EM algorithm.

### 4.1   Fully-Connected Graph Construction

Six types of meta relations {*reply*, *replied-by*, *speak*, *spoken-by*, *address*, *addressed-by*} are introduced in HeterMPC (Gu et al., 2022) to describe directed edges between nodes. However, given an MPC with a few addressee labels missing, existing methods usually return several separate conversation fragments.

To build a consecutively connected conversation graph and ensure message passing between these conversation fragments, additional latent edges are required for completing a fully-connected conversation graph. In this paper, four additional types of latent edges are designed to establish the dependency of utterance nodes on all other nodes in an MPC graph for conversation contextualization as shown in Figure 2. There are two types of them employed to characterize latent relationships between two disconnected utterance nodes. In detail, *latent-reply* characterizes directional edges from latter utterances to previous ones which are ordered by their appearance in an MPC, and vice versa for *latent-replied-by*. On the other hand, another two types of latent edges are employed to characterize the relationships between an utterance node and an interlocutor node. In detail, *latent-address* characterizes directional edges from utterance nodes to interlocutor nodes, and vice versa for *latent-addressed-by*. By this means, both utterances that do not reply to directly and

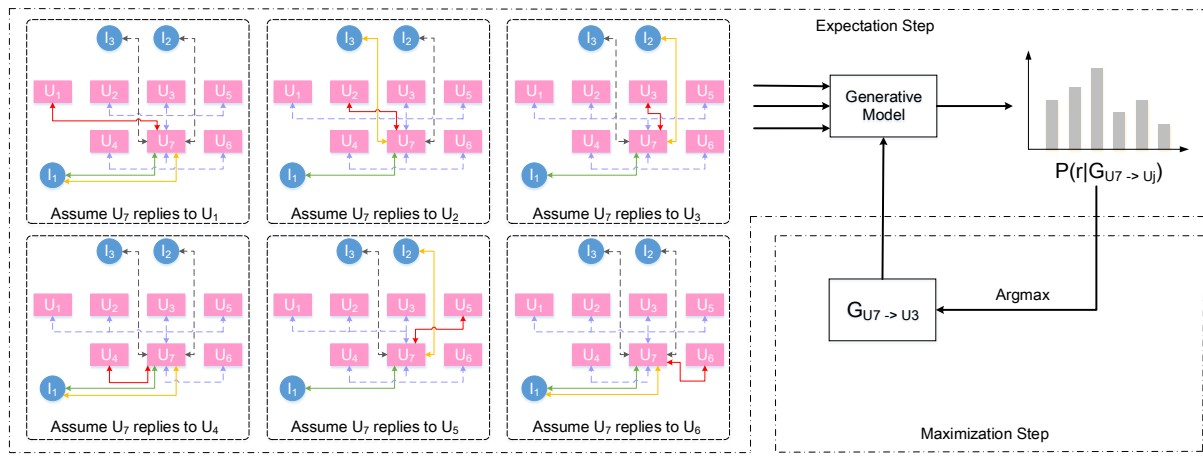

Figure 3: Illustration of the EM training process for the instance in Figure 1, where the expectation and maximization steps are performed alternately.

interlocutors that do not address to directly are also useful for capturing the semantics contained in graph nodes and for modeling complicated interactions between nodes.

## 4.2 EM for Addressee Deduction

Figure 3 illustrates the EM training process, where the expectation and maximization steps are performed alternately. The addressee of an utterance without a golden addressee label is modeled as a discrete latent variable.

**Expectation Step** Given the observed MPC instances with model parameters frozen, the conditional distribution of the latent addressee variable is calculated during the E steps. Here, a specific addressee corresponds to a determined MPC graph. To derive the probability distribution of the latent addressee variable, a set of latent graphs are fed into the generative model respectively to calculate the probabilities of generating a response under these latent graphs as $P(\boldsymbol{r}|\mathbb{G}_{U_i \to U_j}, \boldsymbol{c})$, where $\mathbb{G}_{U_i \to U_j}$ is a graph assuming $U_i$ replies to $U_j$. The derived probability distribution of generating a response under these latent graphs serves as the posterior of selecting the latent variable. The latent addressee distribution of the latent variable is estimated by applying Bayes' rule as:

$$
\begin{aligned}
&P(\mathbb{G}_{U_i \to U_j}|\boldsymbol{c}, \boldsymbol{r}; \boldsymbol{\theta}) \\
&= \frac{P(\boldsymbol{r}|\mathbb{G}_{U_i \to U_j}, \boldsymbol{c}; \boldsymbol{\theta})P(\mathbb{G}_{U_i \to U_j}|\boldsymbol{c}; \boldsymbol{\theta})}{\sum_{k=1}^{i-1} P(\boldsymbol{r}|\mathbb{G}_{U_i \to U_k}, \boldsymbol{c}; \boldsymbol{\theta})P(\mathbb{G}_{U_i \to U_k}|\boldsymbol{c}; \boldsymbol{\theta})}.
\end{aligned}
\tag{2}
$$

A uniform prior for every context $\boldsymbol{c}$ as $P(\mathbb{G}_{U_i \to U_j}|\boldsymbol{c}; \boldsymbol{\theta}) = 1/(i-1)$ is assumed,

which simplifies Eq. (2) as:

$$
P(\mathbb{G}_{U_i \to U_j}|\boldsymbol{c}, \boldsymbol{r}; \boldsymbol{\theta}) = \frac{P(\boldsymbol{r}|\mathbb{G}_{U_i \to U_j}, \boldsymbol{c}; \boldsymbol{\theta})}{\sum_{k=1}^{i-1} P(\boldsymbol{r}|\mathbb{G}_{U_i \to U_k}, \boldsymbol{c}; \boldsymbol{\theta})}.
\tag{3}
$$

**Maximization Step** After deriving the approximate probability distribution of the latent addressee variable, we maximize the expected log-likelihood with respect to $\boldsymbol{\theta}$:

$$
\begin{aligned}
&\mathbb{E}_{\mathbb{G} \sim P(\mathbb{G}_{U_i \to U_j}|\boldsymbol{c}, \boldsymbol{r}; \boldsymbol{\theta})}[\log P(\boldsymbol{r}, \mathbb{G}|\boldsymbol{c}; \boldsymbol{\theta})] \\
&= \sum_{j=1}^{i-1} P(\mathbb{G}_{U_i \to U_j}|\boldsymbol{c}, \boldsymbol{r}; \boldsymbol{\theta}) \log P(\boldsymbol{r}, \mathbb{G}_{U_i \to U_j}|\boldsymbol{c}; \boldsymbol{\theta}).
\end{aligned}
\tag{4}
$$

**Hard EM** A hard EM method (Min et al., 2019) that selects the addressee with the highest probability as the silver label is adopted as:

$$
\bar{U}_j = \underset{U_j}{\operatorname{argmax}} P(\mathbb{G}_{U_i \to U_j}|\boldsymbol{c}, \boldsymbol{r}; \boldsymbol{\theta}), \; j < i, \tag{5}
$$

and the maximization step is approximated as $\log P(\boldsymbol{r}, \mathbb{G}_{U_i \to \bar{U}_j}|\boldsymbol{c}; \boldsymbol{\theta})$. Once the silver addressee label is determined in this round, its corresponding MPC graph is employed for regular training by minimizing the negative log-likelihood loss of responses.

## 4.3 Addressee Initialization

The initialization of addressee labels is crucial to EM for addressee deduction, as it helps converge to optimal model parameters. Thus, an addressee initialization method is designed for utterances without addressee labels before EM training, to select the most probable one as initialization.

The encoder and decoder of the model are initialized with PLMs, e.g., BART, which is pre-trained on texts in the general domain. First, domain adaptation is conducted based on the fully-connected graph constructed in Section 4.1, with the learning objective of minimizing the negative log-likelihood loss of responses on the task training set. By this means, the representation space can be adapted to the task domain and to capture the conversation semantics. Then, for a specific utterance without an addressee label, each previous utterance is assumed to be the replied utterance respectively by setting the corresponding utterance-utterance edges to *reply* and *replied-by*, as well as setting the utterance-interlocutor edges to *address* and *addressed-by*. This process is illustrated in the left part of Figure 3. The probability of generating a response under an assumed graph is calculated following Eq. (2). Finally, the one with the highest probability following Eq. (5) is chosen as the initialization for the first round of M step.

## 5 Experiments

### 5.1 Datasets

We evaluated our proposed methods on two Ubuntu IRC benchmarks. One was released by Ouchi and Tsuboi (2016), in which the addressee labels for part of the history utterances were missing. Here, we adopted the version shared in Le et al. (2019). The conversation sessions were separated into three categories according to the session length (Len-5, Len-10 and Len-15). In this paper, the subset of session length 5 was employed due to the limitation of computing resources. The other dataset where addressee labels were provided for each utterance was adopted following previous work (Hu et al., 2019; Gu et al., 2022). Both datasets were popularly used in the field of multi-party conversations (Zhang et al., 2018; Wang et al., 2020; Gu et al., 2021). Appendix A.1 presents the statistics of the two benchmarks.

### 5.2 Baseline Models

We compared our proposed model with as many MPC generative models as possible. Considering that there are only a few research papers in this field, several recent advanced models were also adapted to provide sufficient comparisons following Gu et al. (2022). Finally, we compared with (1) non-graph-based models including GPT-2 (Radford et al., 2019) and BART (Lewis et al.,

2020), as well as (2) graph-based models including GSN (Hu et al., 2019) and HeterMPC (Gu et al., 2022). Readers can refer to Appendix A.2 for implementation details of these baseline models.

### 5.3 Metrics

To ensure all experimental results were comparable, we used exactly the same automated and human evaluation metrics as those used in previous work (Hu et al., 2019; Gu et al., 2022). Hu et al. (2019) used the evaluation package released by Chen et al. (2015) including BLEU-1 to BLEU-4, METEOR and $ROUGE_L$, which was also used in this paper.[1] Human evaluation was conducted to measure the quality of the generated responses in terms of three independent aspects: 1) relevance, 2) fluency and 3) informativeness. Each judge was asked to give three binary scores for a response, which were further summed up to derive the final score ranging from 0 to 3.

### 5.4 Implementation Details

The corresponding parameters of MADNet followed those in HeterMPC (Gu et al., 2022) for fair comparison. Model parameters were initialized with pre-trained weights of *bart-base* released by Lewis et al. (2020). The AdamW method (Loshchilov and Hutter, 2019) was employed for optimization. The learning rate was initialized as $6.25e\text{-}5$ and was decayed linearly down to 0. The max gradient norm was clipped down to 1.0. The batch size was set to 128 with 2 gradient accumulation steps. The maximum utterance length was set to 50. The number of layers for initializing utterance representations was set to 3, and the number of layers for heterogeneous graph iteration was set to 3. The number of decoder layers was set to 6. The strategy of greedy search was performed for decoding. The maximum length of responses for generation was also set to 50. All experiments were run on a single Tesla A100 GPU. The number of EM iterations was set to 2. The number of epochs in each M step was set to 4, and the learning rate was fixed to $5e\text{-}7$. Each iteration took about 9 and 6 hours for each E step and M step respectively. The validation set was used to select the best model for testing. All code was implemented in the PyTorch framework[2] and is published to help replicate our results.[3]

---

[1]https://github.com/tylin/coco-caption
[2]https://pytorch.org/
[3]https://github.com/lxchtan/HeterMPC

| Metrics / Models | BLEU-1 | BLEU-2 | BLEU-3 | BLEU-4 | METEOR | ROUGE$_L$ |
|---|---|---|---|---|---|---|
| GSN (Hu et al., 2019) | 6.32 | 2.28 | 1.10 | 0.61 | 3.27 | 7.39 |
| GPT-2 (Radford et al., 2019) | 9.12 | 3.40 | 1.93 | 1.39 | 3.28 | 8.92 |
| BART (Lewis et al., 2020) | 11.13 | 3.95 | 2.11 | 1.44 | 4.45 | 10.20 |
| HeterMPC (Gu et al., 2022) | 11.40 | 4.29 | 2.43 | 1.74 | 4.57 | 10.44 |
| MADNet | **11.82$^\dagger$** | **4.58$^\dagger$** | **2.65** | **1.91** | **4.90$^\dagger$** | **10.74$^\dagger$** |
| MADNet w/o. EM for addressee deduction | 11.62 | 4.48 | 2.59 | 1.88 | 4.80 | 10.63 |
| MADNet w/o. latent-reply and latent-replied-by | 11.76 | 4.43 | 2.47 | 1.74 | 4.83 | 10.67 |
| MADNet w/o. latent-address and latent-addressed-by | 11.54 | 4.44 | 2.57 | 1.87 | 4.72 | 10.52 |

Table 1: Evaluation results and ablations on the test set of Ouchi and Tsuboi (2016) in terms of automated evaluation. Numbers in bold denoted that the results achieved the best, and those marked with † denoted that the improvements were statistically significant (t-test with $p$-value $< 0.05$) comparing with the best performing baseline.

| Metrics / Models | BLEU-1 | BLEU-2 | BLEU-3 | BLEU-4 | METEOR | ROUGE$_L$ |
|---|---|---|---|---|---|---|
| GSN (Hu et al., 2019) | 10.23 | 3.57 | 1.70 | 0.97 | 4.10 | 9.91 |
| GPT-2 (Radford et al., 2019) | 10.37 | 3.60 | 1.66 | 0.93 | 4.01 | 9.53 |
| BART (Lewis et al., 2020) | 11.25 | 4.02 | 1.78 | 0.95 | 4.46 | 9.90 |
| HeterMPC (Gu et al., 2022) | 12.26 | 4.80 | 2.42 | 1.49 | 4.94 | 11.20 |
| MADNet | **12.73$^\dagger$** | **5.12$^\dagger$** | **2.64** | **1.63** | **5.31$^\dagger$** | **11.74$^\dagger$** |
| MADNet w/o. latent-reply and latent-replied-by | 12.54 | 4.91 | 2.53 | 1.59 | 5.20 | 11.60 |
| MADNet w/o. latent-address and latent-addressed-by | 12.45 | 4.92 | 2.52 | 1.55 | 5.18 | 11.60 |

Table 2: Evaluation results and ablations on the test set of Hu et al. (2019) in terms of automated evaluation. Results except ours are cited from Gu et al. (2022). Note that EM for addressee deduction was not adopted on this dataset, since addressee labels were provided for each utterance.

## 5.5 Evaluation Results

In our experiments, BART was selected to initialize MADNet following Gu et al. (2022).

**Automated Evaluation** Table 1 and Table 2 present the evaluation results of MADNet and previous methods on the test sets. Each model ran four times with identical architectures and different random initializations, and the best out of them was reported. The results show that MADNet outperformed all baselines in terms of all metrics. Specifically, MADNet outperformed the best performing baseline, i.e., HeterMPC by 0.42% BLEU-1, 0.29% BLEU-2, 0.22% BLEU-3, 0.17% BLEU-4, 0.33% METEOR and 0.30% ROUGE$_L$ on the test set of Ouchi and Tsuboi (2016). Additionally, MADNet outperformed HeterMPC by 0.47% BLEU-1, 0.32% BLEU-2, 0.22% BLEU-3, 0.14% BLEU-4, 0.37% METEOR and 0.54% ROUGE$_L$ on the test set of Hu et al. (2019). These results illustrated the effectiveness of our proposed method in modeling MPC structures, and the importance of message passing between the utterance and interlocutor nodes in an MPC graph.

To further verify the effectiveness of each component of our proposed method, ablation tests were conducted as shown in the last few rows of Table 1 and Table 2. First, EM for addressee deduction was removed on the dataset of Ouchi and Tsuboi (2016). The drop in performance illustrated that accurate addressee labels were crucial to the graphical information flow modeling in MPCs. In addition, EM was an effective solution to addressee deduction. Furthermore, the latent-reply and latent-replied-by edges, or latent-address and latent-addressed-by edges were removed respectively. The drop in performance illustrated the importance of modeling interactions between indirectly related utterances, and those between utterances and interlocutors for better conversation contextualization.

**Human Evaluation** Table 3 presents the human evaluation results on a randomly sampled test set of Ouchi and Tsuboi (2016). 200 samples were evaluated and the order of evaluation systems were shuffled. Three graduate students were asked to score from 0 to 3 (3 for the best) and the average scores were reported. It can be seen that MADNet achieved higher subjective quality scores than the selected baseline models.

| Metrics / Models | Score |
|---|---|
| Human | 2.09 |
| GSN (Hu et al., 2019) | 1.20 |
| BART (Lewis et al., 2020) | 1.54 |
| HeterMPC (Gu et al., 2022) | 1.62 |
| MADNet | 1.79 |

Table 3: Human evaluation results of MADNet and some selected systems on a randomly sampled test set of Ouchi and Tsuboi (2016).

| Metrics / Models | Accuracy | BLEU-4 | METEOR | ROUGE$_L$ |
|---|---|---|---|---|
| HeterMPC | - | 1.33 | 5.03 | 11.35 |
| HeterMPC$_{rand}$ | 37.4 | 1.29 | 4.94 | 11.23 |
| HeterMPC$_{prec}$ | 44.8 | 1.32 | 4.96 | 11.32 |
| MADNet | 50.1 | 1.51 | 5.17 | 11.65 |
| MADNet$_{orac}$ | 100.0 | 1.63 | 5.31 | 11.74 |

Table 4: Accuracy of addressee deduction and automated evaluation results on the modified dataset of Hu et al. (2019). *rand*, *prec* and *orac* were abbreviations of *random*, *preceding* and *oracle* respectively.

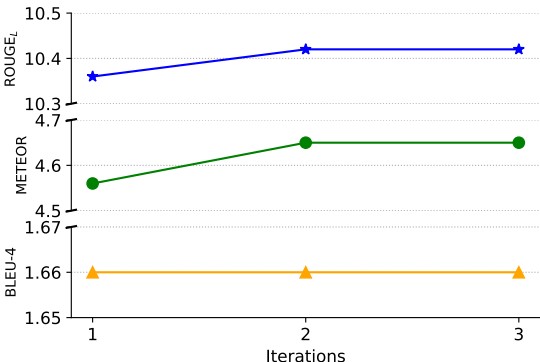

Figure 4: Performance of MADNet under different numbers of EM iterations on the validation set of Ouchi and Tsuboi (2016).

### 5.6 Analysis

**Accuracy of addressee deduction.** A key aspect of the proposed method is to deduce the addressee information which in turn improves the performance of response generation. Therefore, the accuracy of addressee deduction was directly evaluated with respect to a set of baselines to explore its impact on response generation. To do that, a modified dataset of Hu et al. (2019) was constructed. Specifically, the golden addressee label of the last utterance of the conversation history was masked to derive the modified dataset.[4] Results of five selected methods on this modified dataset were compared as shown in Table 4: (1) HeterMPC, (2) each utterance whose addressee label was masked was *randomly assigned a previous utterance* as its reply-to utterance and fed it to HeterMPC, denoted as HeterMPC$_{rand}$, (3) each utterance whose addressee label was masked was *assigned its preceding utterance* as its reply-to utterance and fed it to HeterMPC, denoted as HeterMPC$_{prec}$, (4) MADNet and (5) MADNet with *the oracle addressee labels*, i.e., MADNet on the original Hu et al. (2019) dataset. Results show that the prediction of addressees significantly affects the performance of MPC generation. Seriously wrong predictions might even hurt performance. It can be seen that the addressee deduction with EM in MADNet outperformed the heuristic methods of random selection by a margin of 12.7% accuracy, and of selecting its preceding utterance by a margin of 5.3% accuracy. As a result, the generation performance was improved benefiting from accurate addressee predictions. It is notable that the prediction of addressee achieved only 50.1% accuracy, which shows that this task is still difficult

and there is a lot of room for further improvement.

**Number of EM iterations.** Figure 4 illustrated how the performance of MADNet changed with respect to different numbers of EM iterations on the validation set of Ouchi and Tsuboi (2016). It can be seen that the performance of MADNet was improved as the number of EM iterations increased at the beginning in terms of METEOR and ROUGE$_L$, showing the effectiveness of employing EM for addressee deduction. Then, the performance was stable with more EM iterations. The reason might be that models have selected as many optimal addressee labels as possible.

**Case Study.** A case study was conducted by randomly sampling an MPC instance as shown in Table 5. Given the conversation graph, the response to generate addressed I.1, so the information relevant to I.1 should be collected. It can be seen from this instance that the addressee label was only available for the third utterance, and the established conversation graph was very fragmented due to the lack of addressee labels. Conditioned on an inconsecutively connected graph, previous methods hardly capture the context semantics and can only

---

[4]The dataset of Hu et al. (2019) was adopted here, since the golden addressee labels were provided for each utterance and can be used for evaluation.

| Speaker | Utterance | Addressee |
|---------|-----------|-----------|
| I.1 | perhaps but not everyone uses that | - |
| I.2 | i ll ask him for his history log i think | - |
| I.3 | for people who do n t the phased update percentages are n t considered ok 0 | I.1 |
| I.1 | true | I.3 (Deduced) |
| I.3 | i first thought it might be related to https launchpad net ubuntu source unity scopes api 0 6 19 15 (**Human**) | I.1 |
| | i do n t know how to do that but i m not sure what you want to do with the (**GSN**) | |
| | i m not sure if you can get a silo for that but i m not aware of any other (**BART**) | |
| | i m not sure if you can get that to work for you but i think it s a good (**HeterMPC**) | |
| | i think it s a bit of a corner case for people who do n t have the phased update (**MADNet**) | |

Table 5: The response generation results of a test sample. "I." is an abbreviation of "interlocutor". We kept original texts without manual corrections.

generate generic responses such as "*i m not sure ...*". For MADNet, the missing addressee label of the fourth utterance was deduced as I.3, which was appropriate considering the MPC context. Given the deduced addressee label, the message of "*phased update*" in the third utterance can be passed to the fourth utterance. Furthermore, the response to generate was about to reply to the fourth utterance, and this important message can further captured for response generation.

## 6 Conclusion

We present MADNet to maximize addressee deduction expectation to study the issue of scarcity of addressee labels in multi-party conversations. Four types of latent edges are designed to model interactions between indirectly related utterances, and those between utterances and interlocutors for conversation contextualization. Furthermore, an EM-based approach is designed to deduce silver addressee labels and optimize the quality of generated responses. Experimental results show that the proposed MADNet outperforms previous methods by significant margins on two benchmarks of MPC generation. It especially shows better generalization and robustness in the more common and challenging setting where a few addressee labels are missing.

## Limitations

Although the proposed method has shown great performance to alleviate the scarcity of addressee labels which is a common issue in multi-party conversations, we should realize that the proposed method still can be further improved. For example, to derive the probability distribution of the latent addressee variable, a substitute that the probability of generating a response under the assumed graph is considered as its approximation. This assumption has shown its empirical improvement in our experiments, and the theoretical analysis will be a part of our future work to help derive more accurate probability distribution. In addition, a set of latent graphs are required and fed into the generative model to calculate the probabilities of generating a response under these latent graphs, which consumes much computation resources. Thus, optimization of the expectation steps with less computation is worth studying. Besides, benchmarking the baselines and evaluating the proposed method on other appropriate datasets to make it more representative of as many MPC scenarios as possible will be part of our future work.

## Acknowledgements

This work was supported by the Opening Foundation of State Key Laboratory of Cognitive Intelligence, iFLYTEK COGOS-2022005. We thank anonymous reviewers for their valuable comments.

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

## A  Appendix

### A.1  Datasets

| Datasets | Train | Valid | Test |
|---|---|---|---|
| Ouchi and Tsuboi (2016) | 461,120 | 28,570 | 32,668 |
| Hu et al. (2019) | 311,725 | 5,000 | 5,000 |

Table 6: Statistics of the two benchmarks evaluated in this paper.

### A.2  Baseline Models

We compared our proposed model with as many MPC generative models as possible. Considering that there are only a few research papers in this field, several recent advanced models were also adapted to provide sufficient comparisons. We followed previous work (Hu et al., 2019; Gu et al., 2022) that the tags of speakers and addressees were both used if they were available when establishing the performance of baselines. Finally, we compared with (1) non-graph-based models including GPT-2 (Radford et al., 2019) and BART (Lewis et al., 2020), as well as (2) graph-based models including GSN (Hu et al., 2019) and HeterMPC (Gu et al., 2022) as follows.

**(1) GPT-2** (Radford et al., 2019) was a uni-directional pre-trained language model. Following its original concatenation operation, all context utterances and the response were concatenated with a special [SEP] token as input for encoding. **(2) BART** (Lewis et al., 2020) was a denoising autoencoder using a standard Tranformer-based architecture, trained by corrupting text with an arbitrary noising function and learning to reconstruct the original text. In our experiments, a concatenated context started with  and separated with  were fed into the encoder, and a response were fed into the decoder. **(3) GSN** (Hu et al., 2019) made the first attempt to model an MPC with a homogeneous graph. The core of GSN was an utterance-level graph-structured encoder. **(4) HeterMPC** (Gu et al., 2022) achieved the state-of-the-art performance on MPCs. It proposed to model the complicated interactions between utterances and interlocutors in MPCs with a heterogeneous graph, where two types of graph nodes and six types of edges are designed to model heterogeneity. Two versions of HeterMPC were provided that were initialized with BERT and BART respectively. The latter was adopted in this paper which showed better performance.