# OpenReview forum: "MADNet: Maximizing Addressee Deduction Expectation for Multi-Party Conversation Generation"
_EMNLP/2023/Conference — EMNLP 2023 Main_

### Official Review · Reviewer_rqCf · 2023-08-04

**Soundness:** 4

**Excitement:**

3: Ambivalent: It has merits (e.g., it reports state-of-the-art results, the idea is nice), but there are key weaknesses (e.g., it describes incremental work), and it can significantly benefit from another round of revision. However, I won't object to accepting it if my co-reviewers champion it.

**Paper Topic And Main Contributions:**

The paper presents MADNet, an approach for multi-party conversation generation, addressing the issue of missing addressee labels. The authors propose constructing a fully-connected graph for effective context aggregation, employ an EM algorithm for addressee deduction, and initialize addressees using domain adaptation. The claims are validated empirically and showcase MADNet's superiority in generating responses for multi-party conversations, particularly under scenarios with incomplete addressee information.

**Questions For The Authors:**

* Do you have qualitative examples of responses generated by MADNet in each ablation scenario? Contrasting them with complete responses, would help readers grasp the practical significance of the ablations. Similarly, providing a qualitative analysis / example responses of how the performance evolves with different number of EM iterations would also help.

**Reasons To Accept:**

* Effective Addressee Deduction: The paper's proposes an interesting incorporation of Expectation-Maximization (EM) for accurate addressee deduction. The experiments demonstrate the effectiveness of the approach in both automated evaluation metrics (BLEU, METEOR, ROUGE) and human evaluation.
* Comprehensive Experimental Analysis: The paper's experiments include evaluations on two diverse benchmarks, showcasing the generalizability of their approach. Comparative analysis includes both non-graph-based (GPT-2, BART) and graph-based (GSN, HeterMPC) models, and is strong enough to highlight MADNet's effectivness.
* Ablation Studies: The ablation experiments conducted to analyze the contributions of different model components (e.g., latent-reply, latent-address edges) provide valuable insights into the proposed approach. The authors have done a good job of systematically isolating these components, to demonstrate their individual impact on response generation, which support the modeling choices.
* Human Evaluation and Real-World Applicability: The paper incorporates human evaluation, assessing the quality of generated responses in terms of relevance, fluency, and informativeness.

**Reasons To Reject:**

* Problem Significance and Novelty: The paper failed to establish a clear and compelling case for the significance of the addressed problem, specifically the challenges posed by multi-party conversation context. While the proposed EM-based addressee deduction is innovative, the paper did not thoroughly demonstrate the real-world impact of accurate addressee prediction in enhancing response quality.
* The paper should provide a robust theoretical grounding for the choice of employing EM for addressee deduction. While EM is a well-established optimization technique, its application in the specific context of multi-party conversation graphs requires lacks solid justification. Comparisons with alternative approaches, such as more advanced graph neural networks or hierarchical attention mechanisms, would bolster the methodological choices and theoretical foundation of MADNet.

**Reproducibility:**

4: Could mostly reproduce the results, but there may be some variation because of sample variance or minor variations in their interpretation of the protocol or method.

**Reviewer Confidence:**

4: Quite sure. I tried to check the important points carefully. It's unlikely, though conceivable, that I missed something that should affect my ratings.

---

> ### Author Rebuttal · Authors · 2023-08-28
>
> We thank the reviewers for the valuable suggestions! We are so excited to receive many encouraging reviews and constructive suggestions. We will try our best to respond to the raised issues and will implement the suggested improvements in our revision.
>
>
> 1.Re: **problem significance and novelty**
>
> First, the scarcity of addressee labels is a common issue in multi-party conversations. In detail, as we described in line 69-71, statistics show that addressees of 55% of the utterances in the Ubuntu IRC dataset (Ouchi and Tsuboi, 2016) are not specified.
>
> Second, existing methods rely heavily on the complete and necessary addressee labels, and can only be applied to an ideal setting where each utterance must be tagged with an “@” or other equivalent addressee label. As illustrated in Figure 1, given an MPC with a few addressee labels missing, existing methods fail to build a consecutively connected conversation graph, but only a few separate conversation fragments instead. Therefore, nodes without direct connections cannot exchange information between each other through one-hop message passing. Inevitable information loss and passing latency will affect generation performance significantly. But this common and challenging issue has not been studied in previous work.
>
> Third, empirical experiments in Section 5.4 and case study in Section 5.5 show that existing methods cannot handle the addressed problem. Besides, these results illustrated the effectiveness of our proposed method MADNet in modeling MPC structures and message passing between the utterance and interlocutor nodes in an MPC graph.
>
> In summary, as we highlight at the end of the Section of related work, existing methods target only on an ideal setting where addressee labels of all utterances are necessary, while the proposed method is suitable for more common conversation sessions where a few addressee labels are missing. To the best of our knowledge, this paper makes the first attempt to extend to more common MPC scenarios and to explore the issue of missing addressee labels by maximizing addressee deduction expectation for MPC generation. Thanks for your advice, we will clarify the significance of the problem and the novelty of the method in our revision.
>
>
> 2.Re: **choice of employing EM for addressee deduction**
>
> Addressee labels are an important component of the MPC context, and the empirical experiments show that the missing part of addressee labels will affect the generation performance in turn. Intuitively, it is natural to model the addressee of an utterance without a golden addressee label as a discrete latent variable, and to employ EM for addressee deduction and response generation. In addition, Equations (2)-(5) in Section 4.2 further provide a robust theoretical grounding for the choice of employing EM. Thanks for your advice, comparisons with alternative approaches to addressing the scarcity of addressee labels and to mitigating their effect on MPC generation will be part of our future work.
>
>
> 3.Re: **qualitative examples**
>
> The current submission has presented the quantitative results of various ablation and analysis experiments. Thanks for your advice, more qualitative examples of these corresponding experiments will be included in the revision to help readers better understand the effectiveness of each component of the proposed method.

---

### Official Review · Reviewer_JM8Y · 2023-08-05

**Soundness:** 4

**Excitement:**

4: Strong: This paper deepens the understanding of some phenomenon or lowers the barriers to an existing research direction.

**Paper Topic And Main Contributions:**

This paper proposes a new method for generating multi-party conversations using graph neural networks, even when addressee labels are missing. By maximizing addressee deduction expectation and designing additional latent edges, MADNet outperforms various baseline models. It uses an Expectation-Maximization (EM) algorithm to iteratively generate silver addressee labels and optimize the quality of generated responses. In particular, the paper uses a Hard EM method that selects the addressee with the highest probability as the silver label. The maximization step is approximated as log P (r, GU_i→U_j|c; θ). Once the silver addressee label is determined in this round, its corresponding MPC graph is employed for regular training by minimizing the negative log-likelihood loss of responses

**Questions For The Authors:**

1. Have you considered any other evaluation metrics besides BLEU, METEOR, and ROUGE-L, which are commonly used in the literature?
2. Can you provide more details on the computational complexity of the proposed method and how it scales with the size of the input data?
3. Can you try to use LLMs in comparison to your work?

**Reasons To Accept:**

1. A new method called MADNet for generating multi-party conversations using graph neural networks, even when addressee labels are missing.
2. A novel approach to maximizing addressee deduction expectation by designing additional latent edges in the graph.
3. An extensive evaluation of MADNet on the Ubuntu IRC channel benchmarks, showing that it outperforms various baseline models in terms of both automatic and human evaluations.
4. A detailed analysis of the impact of different design choices on the performance of MADNet, providing insights for future research in this area.

**Reasons To Reject:**

1. The paper only evaluates the proposed method on two benchmarks, which may not be representative of all possible multi-party conversation scenarios.
2. The paper does not provide a detailed analysis of the computational complexity of the proposed method, which may be a concern for large-scale applications.

**Reproducibility:**

4: Could mostly reproduce the results, but there may be some variation because of sample variance or minor variations in their interpretation of the protocol or method.

**Reviewer Confidence:**

3: Pretty sure, but there's a chance I missed something. Although I have a good feel for this area in general, I did not carefully check the paper's details, e.g., the math, experimental design, or novelty.

---

> ### Author Rebuttal · Authors · 2023-08-28
>
> We thank the reviewers for the valuable suggestions! We are so excited to receive many encouraging reviews and constructive suggestions. We will try our best to respond to the raised issues and will implement the suggested improvements in our revision.
>
>
> 1.Re: **choice of datasets**
>
> As we described in Section 5.1, to ensure all experimental results were comparable, we used exactly the same dataset as those used in previous work (Hu et al., 2019; Gu et al., 2022), and an additional Ubuntu-based dataset (Ouchi and Tsuboi, 2016) in which the addressee labels for part of the history utterances were missing. These datasets were both popularly used in the field of multi-party conversations (Ouchi and Tsuboi, 2016; Zhang et al., 2018; Le et al., 2019; Wang et al., 2020; Gu et al., 2021). Furthermore, as we described in line 69-71, statistics show that addressees of 55% of the utterances in the Ubuntu IRC dataset (Ouchi and Tsuboi, 2016) are not specified. These statistics illustrate that the scarcity of addressee labels is a common issue, thus the Ouchi and Tsuboi (2016) dataset is very suitable for the experimental setting studied in this paper. Thanks for your advice, benchmarking the baselines and evaluating the proposed method on other appropriate datasets to make it more representative of as many multi-party conversation scenarios as possible will be part of our future work.
>
>
> 2.Re: **computational complexity**
>
> As we discussed in the section of Limitation, we have been aware that computational complexity optimization of the proposed method is indeed an issue worth further investigation. In detail, a set of latent graphs, each of which assumes the current utterance replies to one of the utterances in the previous conversation history as shown in Figure 3, are required and fed into the generative model to calculate the probabilities of generating a response under these latent graphs. This process typically consumes much computation resources. Thanks for your advice, more analysis of computational complexity will be included in the revision.
>
>
> 3.Re: **evaluation metrics**
>
> Thanks for your advice. As we described in Section 5.3, to ensure all experimental results were comparable, we used exactly the same automated and human evaluation metrics as those used in previous work (Hu et al., 2019; Gu et al., 2022). Evaluating the proposed method and various baselines using more other automated evaluation metrics might shed light on comprehensive evaluation, which will be part of our future work.
>
>
> 4.Re: **comparison with LLM**
>
> Thanks for your advice. In fact, we have already studied the application of LLMs to multi-party conversations, which aims at evaluating if LLMs show great performance on this task as comprehensively as possible. Some preliminary results and conclusions have been obtained but not yet finalized. This part of work is still in progress, and hopefully will be released to the community soon.

---

### Official Review · Reviewer_wap4 · 2023-08-10

**Soundness:** 4

**Excitement:**

3: Ambivalent: It has merits (e.g., it reports state-of-the-art results, the idea is nice), but there are key weaknesses (e.g., it describes incremental work), and it can significantly benefit from another round of revision. However, I won't object to accepting it if my co-reviewers champion it.

**Paper Topic And Main Contributions:**

This paper makes the first attempt to explore the issue of missing addressee labels and to target on more common MPC scenarios. They proposed a fully-connected heterogeneous graph architecture along with EM training is designed to help deduce the addressee of an utterance for improving MPC generation. Experimental results show that the proposed model achieves a new state-of-the-art performance of MPC generation on two Ubuntu IRC benchmarks.

**Questions For The Authors:**

1.Could you provide details examples of the address labels and show some good case and bad case?
2.why choose ubuntu dataset for experiments. I think it may be not a proper dataset for address labels tasks. Do you have any statistics for the address labels in ubuntu and other datasets?
3. address labels could be various, e.g.  city, street, etc.  Do you consider this difference?

**Reasons To Accept:**

Despite its importance, address labels are missing in the existing multi-turn conversation datasets, which limited the further research progress in this track. This paper provide a method to augment the address labels in the generated datasets, and proved the effectiveness. The paper is well organized and easy to follow.

**Reasons To Reject:**

There could be more analysis to the address labels with case study. .ubuntu may be not a proper dataset for address labels tasks.

**Reproducibility:**

3: Could reproduce the results with some difficulty. The settings of parameters are underspecified or subjectively determined; the training/evaluation data are not widely available.

**Reviewer Confidence:**

4: Quite sure. I tried to check the important points carefully. It's unlikely, though conceivable, that I missed something that should affect my ratings.

---

> ### Author Rebuttal · Authors · 2023-08-28
>
> We thank the reviewers for the valuable suggestions! We are so excited to receive many encouraging reviews and constructive suggestions. We will try our best to respond to the raised issues and will implement the suggested improvements in our revision.
>
>
> 1.Re: **analysis of addressee labels**
>
> Quantitative and qualitative analysis of addressee labels were conducted in Section 5.5.
>
> In detail, the quantitative accuracy of addressee deduction was directly evaluated with respect to a set of baselines to explore its impact on response generation. Results show that the prediction of addressees significantly affects the performance of MPC generation. It can be seen that the addressee deduction with EM in MADNet outperformed the heuristic methods. As a result, the generation performance was improved benefiting from accurate addressee predictions.
>
> On the other hand, the qualitative case study was conducted as shown in Table 5. It can be seen that the established conversation graph was very fragmented due to the lack of addressee labels. Conditioned on an inconsecutively connected graph, previous methods hardly capture the context semantics and can only generate generic responses. For MADNet, the missing addressee label can be appropriately deduced considering the MPC context. Given the deduced addressee label, some important message can be passed accordingly and further be captured for response generation.
>
> Thanks for your advice, more analysis and examples of addressee labels will be included in the revision.
>
>
> 2.Re: **choice of datasets**
>
> As we described in Section 5.1, to ensure all experimental results were comparable, we used exactly the same dataset as those used in previous work (Hu et al., 2019; Gu et al., 2022), and an additional Ubuntu-based dataset (Ouchi and Tsuboi, 2016) in which the addressee labels for part of the history utterances were missing. These datasets were both popularly used in the field of multi-party conversations (Ouchi and Tsuboi, 2016; Zhang et al., 2018; Le et al., 2019; Wang et al., 2020; Gu et al., 2021). Furthermore, as we described in line 69-71, statistics show that addressees of 55% of the utterances in the Ubuntu IRC dataset (Ouchi and Tsuboi, 2016) are not specified. These statistics illustrate that the scarcity of addressee labels is a common issue, thus the Ouchi and Tsuboi (2016) dataset is very suitable for the experimental setting studied in this paper. Thanks for your advice, benchmarking the baselines and evaluating the proposed method on other appropriate datasets will be part of our future work.
>
>
> 3.Re: **difference of addressee labels**
>
> It is necessary to clarify that addressee labels mentioned in this paper do not mean the locations such as city, street, etc. As we described in Figure 1(a) which is a real-world conversation example in the Ubuntu dataset and Table 5 which presents a case study, addressee labels are mostly showcased by the “@” tags. Thus addressee are typically the interlocutors that appear in the previous conversation history.

---

### Official Review · Reviewer_dY2D · 2023-08-11

**Soundness:** 3

**Excitement:**

4: Strong: This paper deepens the understanding of some phenomenon or lowers the barriers to an existing research direction.

**Paper Topic And Main Contributions:**

This paper is addressing the problem of missing addressee labels in multi-party conversation (MPC) datasets. In MPCs, each utterance is addressed to a specific person, but often the addressee labels indicating who each utterance is addressing are missing in the datasets. The authors propose a model called MADNet that estimates the missing addressee labels using an Expectation-Maximization (EM) approach. They also add latent edges between utterances and interlocutors to keep the conversation graph fully connected even with missing labels.

**Reasons To Accept:**

* They address an interesting problem - missing addressee labels are common in real multi-party conversation datasets, but existing models require complete labels. This work makes models more robust.
* The proposed solution is methodically sound and innovative - using Expectation-Maximization to estimate missing labels is a clever application of a standard technique.


**Reasons To Reject:**

1. They only experiment on Ubuntu IRC datasets. Testing on more diverse MPC datasets would strengthen the generality of the results.
3. The computational overhead of running Expectation-Maximization during training is not discussed in detail. This could be a practical limitation.
5. In Table 1 and 2, MADNET w/o the EM algorithm already performs quite well -- so, is the use of EM algorithm really helping?
1. Human Evaluation is totally wrong.
    1. What questions were asked to the user for human evaluation, what each of the levels in the scale meant. The UI used for human evaluation also needs to be shared. Without these details, human evaluation numbers are all useless.
    1. Don't average different aspects.
    2. Don't use 0-1 scale. Use a likert scale with more levels and proper meaning for each level.

**Reproducibility:**

4: Could mostly reproduce the results, but there may be some variation because of sample variance or minor variations in their interpretation of the protocol or method.

**Reviewer Confidence:**

4: Quite sure. I tried to check the important points carefully. It's unlikely, though conceivable, that I missed something that should affect my ratings.

---

> ### Author Rebuttal · Authors · 2023-08-28
>
> We thank the reviewers for the valuable suggestions! We are so excited to receive many encouraging reviews and constructive suggestions. We will try our best to respond to the raised issues and will implement the suggested improvements in our revision.
>
>
> 1.Re: **choice of datasets**
>
> As we described in Section 5.1, to ensure all experimental results were comparable, we used exactly the same dataset as those used in previous work (Hu et al., 2019; Gu et al., 2022), and an additional Ubuntu-based dataset (Ouchi and Tsuboi, 2016) in which the addressee labels for part of the history utterances were missing. These datasets were both popularly used in the field of multi-party conversations (Ouchi and Tsuboi, 2016; Zhang et al., 2018; Le et al., 2019; Wang et al., 2020; Gu et al., 2021). Furthermore, as we described in line 69-71, statistics show that addressees of 55% of the utterances in the Ubuntu IRC dataset (Ouchi and Tsuboi, 2016) are not specified. These statistics illustrate that the scarcity of addressee labels is a common issue, thus the Ouchi and Tsuboi (2016) dataset is very suitable for the experimental setting studied in this paper. Thanks for your advice, benchmarking the baselines and evaluating the proposed method on other appropriate datasets to make it more representative of as many multi-party conversation scenarios as possible will be part of our future work.
>
>
> 2.Re: **computational overhead**
>
> As we discussed in the section of Limitation, we have been aware that computational complexity optimization of the proposed method is indeed an issue worth further investigation. In detail, a set of latent graphs, each of which assumes the current utterance replies to one of the utterances in the previous conversation history as shown in Figure 3, are required and fed into the generative model to calculate the probabilities of generating a response under these latent graphs. This process typically consumes much computation resources. Thanks for your advice, more analysis of computational complexity will be included in the revision.
>
>
> 3.Re: **performance of using EM**
>
> On the one hand, as shown in Table 1, after removing EM for addressee deduction, the drop in performance illustrated that accurate addressee labels were crucial to the graphical information flow modeling in MPCs. Thus, EM was an effective solution to addressee deduction.
>
> On the other hand, as shown in Table 4, the accuracy of addressee deduction was directly evaluated with respect to a set of baselines to explore its impact on response generation. Results show that the prediction of addressees significantly affects the performance of MPC generation. It can be seen that the addressee deduction with EM in MADNet outperformed the heuristic methods. As a result, the generation performance was improved benefiting from accurate addressee predictions with EM.
>
> Thus, the use of EM algorithm really works for improving the MPC generation performance.
>
>
> 4.Re: **human evaluation**
>
> As we described in Section 5.3, to ensure all experimental results were comparable, we used exactly the same automated and human evaluation metrics as those used in previous work (Hu et al., 2019; Gu et al., 2022). Thanks for your advice, more details of human evaluation will be included in the revision. Besides, evaluating the proposed method and various baselines using an improved human evaluation protocol for more comprehensive evaluation will be part of our future work.

---

### Official Review · Reviewer_SdPB · 2023-08-11

**Soundness:** 4

**Excitement:**

4: Strong: This paper deepens the understanding of some phenomenon or lowers the barriers to an existing research direction.

**Paper Topic And Main Contributions:**

This paper proposes MADNet, a graph-neural-network-based multi-party conversation generation model that is applicable in more realistic settings where addressee labels of utterances can be missing. Previous models require the presence of addressee labels for all utterances.
The contributions are as follows:
1. The authors proposed MADNet which achieved SOTA results on two existing datasets.
2. The authors explored a more realistic setting for multi-party conversation generation where some addressee labels of utterances are missing
3. The authors proposed to use EM training to deduce the addressee of an utterance, which is approved effective.

**Questions For The Authors:**

Question A: The graph neural network architecture is not clear enough. Can you provide more details of your Graph Neural Network model (for example, using math formulas to explain the exact computational steps)?

Question B: How many utterances are there without addressee labels in the dataset released by Ouchi and Tsuboi (2016)?

**Reasons To Accept:**

1. MADNet achieves SOTA results in terms of both automatic and human evaluation.
2. The proposed EM method for addressee deduction is interesting and effective.
3. The paper is well written, the contributions are clearly described, and the model choices are well supported by ablation studies.

**Reasons To Reject:**

No major weaknesses. The only concern is the reliability of evaluation metrics. In conversation generation, there can be multiple "good answers" which have very low lexical overlap (all automatic evaluation metrics used in the paper are based on lexical overlap.) with the ground truth. In this case a low score may not necessarily indicate a bad generation. Human evaluation on the other hand is a more reliable method. The authors need to report inter-annotator agreement to show the human evaluation results are robust and consistent.

**Reproducibility:**

3: Could reproduce the results with some difficulty. The settings of parameters are underspecified or subjectively determined; the training/evaluation data are not widely available.

**Reviewer Confidence:**

3: Pretty sure, but there's a chance I missed something. Although I have a good feel for this area in general, I did not carefully check the paper's details, e.g., the math, experimental design, or novelty.

---

> ### Author Rebuttal · Authors · 2023-08-28
>
> We thank the reviewers for the valuable suggestions! We are so excited to receive many encouraging reviews and constructive suggestions. We will try our best to respond to the raised issues and will implement the suggested improvements in our revision.
>
>
> 1.Re: **reliability of evaluation metric**
>
> Thanks for your advice. As we described in Section 5.3, to ensure all experimental results were comparable, we used exactly the same automated and human evaluation metrics as those used in previous work (Hu et al., 2019; Gu et al., 2022). More detailed human evaluation as you suggest such as inter-annotator agreement might provide more comprehensive evaluation towards “multiple good answers”, which will be part of our future work.
>
>
> 2.Re: **more details of the graph neural network model**
>
> Due to page limitations, we briefly describe in Section 3 the key process of graph construction, node initialization, node updating, and decoder of the baseline, i.e., HeterMPC (Gu et al., 2022), which shares the graph neural network backbone with our method. In order to avoid lengthy method descriptions and to help readers quickly understand the contributions of our proposed method, only some key computational steps are present in this paper. Thanks for your advice, we will describe more details of the graph neural network model as much as the number of pages allows. Readers can also refer to HeterMPC (Gu et al., 2022) for more details.
>
>
> 3.Re: **number of utterances without addressee labels in the Ouchi and Tsuboi (2016) dataset**
>
> As we described in line 69-71, statistics show that addressees of 55% of the utterances in the Ubuntu IRC dataset (Ouchi and Tsuboi, 2016) are not specified, illustrating that the scarcity of addressee labels is a common issue.

---

### Meta-Review · Area_Chair_eJ4L · 2023-09-18

**Recommendation:** 4

**Metareview:**

The paper presents a multi-party conversation generation model that is speaker-aware and works when speaker annotation is missing.

Reviewers appreciated the realistic setting where speaker labels could be missing. There was strong consensus on the soundness of the paper.

Some of the critical suggestions include:
- Better details on the human evaluation setup.
- Improved description of the graph-neural network.
- Enhanced discussion on complexity analysis.
- Baselines suggested by reviewers.

These changes should be incorporated in the revision as promised by the authors.

---

### Decision · Program_Chairs · 2023-10-07

**Decision:**

Accept-Main

**Comment:**

The paper presents a multi-party conversation generation model that is speaker-aware and works when speaker annotation is missing.

Reviewers appreciated the realistic setting where speaker labels could be missing. There was strong consensus on the soundness of the paper.

Some of the critical suggestions include:
- Better details on the human evaluation setup.
- Improved description of the graph-neural network.
- Enhanced discussion on complexity analysis.
- Baselines suggested by reviewers.

These changes should be incorporated in the revision as promised by the authors.